# Effects of Artificially Induced Leg Length Discrepancy on Treadmill-Based Walking and Running Symmetry in Healthy College Students: A Lab-Based Experimental Study

**DOI:** 10.3390/s23249695

**Published:** 2023-12-08

**Authors:** Maria Korontzi, Ioannis Kafetzakis, Dimitris Mandalidis

**Affiliations:** Sports Physical Therapy Laboratory, Department of Physical Education and Sports Science, School of Physical Education and Sports Science, National and Kapodistrian University of Athens, 17237 Athens, Greece; marikoro@phed.uoa.gr (M.K.); kafetzos@phed.uoa.gr (I.K.)

**Keywords:** leg length inequality, gait analysis, walking, running, postural balance

## Abstract

Leg length discrepancy (LLD) is a common postural deviation of musculoskeletal origin, which causes compensatory reactions and often leads to injury. The aim of the study was to investigate the effect of artificially induced LLD on gait symmetry by means of the spatiotemporal gait parameters and ground reaction forces (GRFs) using a treadmill equipped with capacitive sensors (instrumented) as well as the EMG activity of trunk and hip muscles during walking and running. Twenty-six healthy male and female college students were required to perform two sets of four 2.5-min walking and running trials on an instrumented treadmill at 5.6 and 8.1 km·h^−1^, respectively, without (0) and with 1, 2, and 3 cm LLD implemented by wearing a special rubber shoe. Statistical analysis was performed using one-way repeated measures or a mixed-design ANOVA. Most spatiotemporal gait parameters and GRFs demonstrated an increase or decrease as LLD increased either on the short-limb or the long-limb side, with changes becoming more apparent at ≥1 cm LLD during walking and ≥2 cm LLD during running. The EMG activity of trunk and hip muscles was not affected by LLD. Our findings showed that gait symmetry in terms of treadmill-based spatiotemporal parameters of gait and GRFs is affected by LLD, the magnitude of which depends on the speed of locomotion.

## 1. Introduction

Leg length discrepancy (LLD) refers to a skeletal, or sometimes functional, condition occurring in most of the population in which the length between the lower limbs appears to be different [1,2]. A systematic review assessing the prevalence of skeletally induced LLD by radiographic measurements revealed that 90% of the normal population had some type of leg bone length variation, with 20% showing a difference of >0.9 cm [1]. In many people, LLD may go unnoticed during their lifetime [3], but quite often it is detected as an incidental finding in various musculoskeletal pathologies (e.g., osteoarthritis [4,5]) and injuries of the lower extremities (e.g., stress fractures, plantar fasciitis [6,7]) as well as in pain syndromes (e.g., low back pain [8,9]). The asymmetric loading of bones, joints, and soft tissue structures [5,10] caused by deviation of the mechanical axes of the lower limbs and the resulting compensatory postural responses, such as pelvis obliquity [11,12] and torsion of the innominate bones [13], as well as lumbar scoliosis and hypo- or hyperlordosis [9,14,15], that occur due to LLD are thought to be largely responsible for these problems. Research has shown that the significant although controversial effects of postural as well as kinematic and kinetic compensatory responses induced by LLD on body function [16,17,18,19,20] have led authors to question the magnitude of LLD that should be considered clinically significant. An extensive systematic review of the existing literature revealed that while some researchers state that structural LLD as low as 1.0 cm may need to be corrected as it can cause complications in the human body in the long term (e.g., low back pain [21], plantar fasciitis [6], knee osteoarthritis [22]), others report that differences between leg lengths of up to 6.0 cm have no significant effect on human function (e.g., ground reaction forces, energy consumption) [23].

Among the compensatory responses that may be exhibited in individuals with LLD are those related to the spatiotemporal and dynamic parameters of gait during walking [19,24,25]. Individuals with LLD tend to acquire an asymmetrical gait [19,26] by developing a limp or an asymmetric step that will eventually distribute body weight and joint stress unevenly. In this context, some authors have reported that the short leg is subject to greater forces [16,17,18,20,27,28], while others have suggested that the forces exerted are greater on the longer leg [19,29,30], providing, however, limited information on the distribution of forces between the plantar surfaces of the foot [28]. These responses may not only lead to an increased energy expenditure [31], but also to joint problems over time (e.g., arthritis) [4,5]. Other authors have investigated the impact of structural or artificially induced LLD on the muscles that control the pelvis and the trunk in the frontal plane such as the abductors of the hip (e.g., gluteus medius) and/or the extensors of the trunk (e.g., erector spinae) [24,31,32]. These muscles were expected to modify their activity bilaterally by increasing or decreasing their EMG activity to counteract the postural compensatory responses that occur due to LLD [33]. Such muscular imbalances in the lumbopelvic region have been associated with skeletal asymmetry and the instability of adjacent joints [2,34,35] that can potentially cause lumbar discomfort and lead to lower extremity injuries and/or pain [2,36,37]. These effects may be more detrimental in athletes, especially runners, as the loads exerted on the lower extremities are expected to be three times greater compared to walking [38]. However, relevant studies are absent, preventing the implementation of appropriate measures to reduce the adverse effects of LLD on the performance and musculoskeletal integrity of athletes.

The lack of sufficient information regarding the effects of LLD on human function under more demanding conditions, as in running, can partly be explained by the fact that these effects have been studied so far during overground walking [19,24,27,29,31]. Investigating the effects of LLD in this way may prevent standardization of experimental protocols and the study of parameters such as EMG activity, especially if appropriate equipment is not available (e.g., wireless EMG data transmitters). Treadmills, on the other hand, enable clinicians to study walking and running on-site, thereby allowing them to standardize gait conditions and variables such as speed, incline, and test duration, as well as to record muscle activity with the necessary control. More data related to the spatiotemporal characteristics of gait as well as ground reaction forces (GRFs) can also be collected from treadmills if they are instrumented, i.e., treadmills equipped with capacitive sensors, which are embedded beneath the running belt. Moreover, the use of such equipment may not only provide the opportunity to detect LLD but also to control the therapeutic interventions used to treat it, as has been carried out by other means (e.g., motion capture system [39,40]), since it is expected to provide highly reproducible data. The purpose, therefore, of this study was to expand the existing knowledge of the potential effects of LLD on gait symmetry in terms of spatiotemporal parameters and GRFs exertion as well as the EMG activity of muscles acting on the lumbopelvic region during walking and running using an instrumented treadmill.

## 2. Materials and Methods

### 2.1. Sample

A mixed-design approach was utilized to explore the impact of artificially induced leg length discrepancies (LLD) of 0, (no-LLD), 1, 2, and 3 cm (within-subject factor) on the long-limb and short-limb sides (LLS and SLS, between-subjects factor) in 26 male (*n* = 15) and female (*n* = 11) uninjured, physically active (≥2–3 exercise sessions/week of moderate intensity) college students during walking and running. The participants had an average age of 21.2 ± 3.0 years, a height of 1.7 ± 0.1 m, a body mass of 62.7 ± 10.0 kg, and a BMI of 21.8 ± 2.5 kg·m^−2^.

Participants were recruited from the Department of Physical Education and Sports Science of the local state university based on their availability and willingness to participate and each of them signed an informed written consent prior to testing. Data were collected at the university’s physical therapy facilities between March and July of 2022 following ethical approval, which was obtained from the University’s Human Research Ethics Committee (1303/14 July 2021). The size of the study sample was considered adequate to achieve statistical significance with a = 0.05, with an 80% power and effect size (f) = 0.25 (calculated based on a partial η^2^ = 0.06), based on an a priori power analysis using an online power analysis application (G*Power v. 3.1.9.2; Universität Düsseldorf: Psychologie – HHU, Düsseldorf, Germany). Individuals with musculoskeletal soft tissue or joint injuries of the lower limb, pathological conditions that could prevent normal walking and running, or a leg length discrepancy of >5 mm, as well as excessive skeletal deviations in the trunk (e.g., >5.0° of rotation of the thoracic or lumbar spine when performing the Adam’s test indicative of ≥20° scoliosis), pelvis (e.g., >2° obliquity [41]), and feet (e.g., overpronation or supination as defined by the Foot Posture Index-6, (FPI-6 [42])), were excluded from the study. All inclusion/exclusion criteria that were related to prior lower extremity injuries and frontal plane body deviations were selected based on the clinical applicability and the potential impact that they could have on the study variables. Both interviews and musculoskeletal assessment were performed by the first author (M.K.).

### 2.2. Musculoskeletal Assessment

Leg length was measured with a tape measure from the anterior superior iliac crest to the lower rim of the medial malleolus of the ipsilateral lower limb, with each subject supine on the treatment table. Scoliosis was determined by measuring trunk rotation in the thoracic and lumbar spine by placing a bubble scoliometer at the top of the rib hump perpendicular to the long axis of the body at the corresponding spine curves with the trunk flexed (Adam’s test) [43]. Foot type was determined with the FPI-6, a six-item index used for quantifying an individual’s resting standing foot posture by grading (i) talar head position with palpation, as well as (ii) the curves above and below the lateral malleoli, (iii) the calcaneal angle, (iv) the talonavicular bulge, (v) the medial longitudinal arch congruence, and (vi) the forefoot abduction/adduction with inspection, with each subject standing upright in bipedal support with his/her gaze at a fixed point ahead. The obliquity of the pelvis was determined trigonometrically by calculating the angle formed between the line joining the right and left anterior superior iliac spines (inter-ASISs distance) and the horizontal plane. The distances were measured with each subject in the upright standing posture using two laser technology distance meters (PLR 50, Robert Bosch GmbH, Leinfelden-Echterdingen, Germany) and a tape measure adapted to a specially constructed device for measuring skeletal deviations [44]. All measurements, having been deemed reliable for clinical use [42,45,46], were performed twice and their average was considered in the analysis. The participants’ physical activity level was determined based on the Greek version [47] of the modified Baecke questionnaire for usual physical activity [48]. 

### 2.3. Testing Procedure

The study was conducted in two sessions. During the first session, volunteers provided information about their past/present medical history and underwent a thorough clinical assessment; based on this, they were selected (or not) to participate in the study. Only subjects who met the inclusion criteria participated in the second session where the EMG activity of the thoracic erector spinae (TES), lumbar erector spinae (LES), and gluteus medius (GLU), along with the spatiotemporal and dynamic gait parameters, were recorded simultaneously by synchronizing a biosignal recording device and a plantar pressure platform embedded on a treadmill (Figure 1). All tests were selected based on the research question and specific objectives of the study, the validity and reliability of the tests applied, their use by researchers in previous studies that investigated similar research questions, their practicality and feasibility in terms of resources required, the time constraints of the study, their ease of implementation, and their appropriateness in relation to whether they cause undue harm or inconvenience.

Participants were required to perform two sets of four 2.5-min treadmill-based walking and running tests at 5.6 and 8.1 km·h^−1^, respectively, without (0) and with artificially induced LLD of 1, 2, and 3 cm wearing a special rubber shoe lift (EVEN Up, OPED, Oberlaindern, Germany). Both speeds were chosen among those recommended for physical activities in healthy active individuals. The speed of 5.6 km·h^−1^ (average brisk walking speed) was chosen as it is recommended by the Centers for Disease Control and Prevention (CDC) as an adequate walking speed for health-promoting physical activity in adults [49]. The speed of 8.1 km·h^−1^ corresponded to the so-called average slow jogging speed [50] and was chosen as it was expected to reduce the possibility of participants falling, especially when running with 3 cm LLD creating at the same time less EMG artifacts. The special shoes used in the present study to artificially induce LLD are recommended to even up the length of the legs in individuals who are required to wear a walking boot as a means of treatment. They were expected to allow participants to walk and run with ease and stability as they fitted externally to the shoe they wore and were secured with Velcro straps (Figure 2).

Each person had to wear these special shoes on both feet, the size of which was chosen based on the size of the shoe each person normally wears for running, with the sole of one shoe being 1, 2, or 3 cm thicker than the sole of the contralateral shoe. The thickness of the sole of the shoe was increased by placing additional light rubber insoles of a thickness similar to that intended to cause LLD. To prevent fatigue effect on the test results, the order of the testing conditions (magnitude of LLD-by-speed of locomotion) were randomized. Each participant performed adequate practice prior to testing to become familiar with walking and running in the special shoe each time the sole thickness of the shoe was changed. To avoid a lack of postural control, each participant rested between conditions as long as required to achieve heart rate (HR) recovery to 60% of their maximum HR [51].

### 2.4. Spatiotemporal and Ground Reaction Forces Measurements

An instrumented treadmill (Pluto^®^ Med, h/p/cosmos^®^ Sports & Medical GmbH, Nussdorf–Traunstein, Germany) with a pressure platform (FDM-THPL-M-3i, Zebris Medical GmbH, Isny, Germany) consisting of 7168 capacitive sensors embedded beneath the running belt was used to record spatiotemporal data (e.g., step/stride length, step/stride and stance/swing time, stride width, cadence) and GRFs exerted on the plantar rearfoot, midfoot, and forefoot areas, as have been geometrically determined by the manufacturer from the rear 30%, mid 30%, and fore 40% area of a footprint, respectively, during level walking and running at a sampling rate of 240 Hz.

### 2.5. EMG Activity Measurements

The EMG activity of the TES, LES, and GLU was recorded with a biosignal recording system (Biopac MP 100, Aero Camino, Goleta, CA, USA), which consists of a differential amplifier and six input channels, by placing the device bilaterally in the direction of muscle fibers self-adhesive unipolar Ag-AgCl electrodes of 1.0 cm in diameter with an inter-electrode distance of 2 cm. Pairs of electrodes were placed 5 cm lateral to the T9 spinous process for TES [52], 2.5 cm lateral to the L1 spinous process for LES [53], and at 50% of the distance from the apex of the iliac crest to the greater trochanter for GLU [54]. The ground electrode was placed on the clavicle. Electrode placement sites on the skin for the erector spinae and gluteus medius were located by palpation with each subject in standing and side lying positions, respectively. Where necessary, the skin surface was prepared by removing hair and was cleaned with ethyl alcohol for better signal transmission.

#### 2.5.1. Recording, Processing, and Analysis of the Signal

The EMG activity recording frequency was 1000 samples/sec using a 10–500 Hz Band Pass FIR filter, while a 50 Hz Band Stop FIR filter was applied to limit noise from the electric current. The visualization, storage, and analysis of the data were performed using special software (AcqKnowledge^®^, v. 3.9.1.6, Biopac Systems, Inc. Aero Camino, Goleta, CA, USA) on a computer to which the data were transmitted by wire in real time. The EMG activity of the TES, LES, and GLU in each condition of the experimental procedure was determined based on the calculation of the root mean square (RMS) in mV using a window of 30 msec and expressed as a percentage of each muscle’s maximum voluntary isometric contraction (MVIC). Statistical analysis was performed on the mean EMG activity recorded during 150 s of treadmill walking or running, as determined by the first and last contact applied by the foot on its first and last step, respectively, in each of the conditions tested (Figure 1).

#### 2.5.2. Normalization of the EMG Activity

The maximum voluntary isometric contractions (MVICs) of the muscles under investigation, the generated activity of which was used to normalize the muscle activity in each experimental condition, were performed before the start of the research protocol. The MVICs of the TES and LES were performed against resistance provided by a belt placed around the upper thoracic torso (middle of the scapulae) with each participant in the prone position. The MIVCs of the right and left GLU were performed against resistance provided by a belt placed around the lateral lower third of the thigh just above the knee joint, with the lower limb horizontally aligned using a Saunders digital inclinometer (The Saunders Group Inc., Chaska, MN, USA) and the knees fully extended with each participant in the side lying position. Additional belts around the pelvis and the lower limbs were used to stabilize the participants on the treatment table. Three 5-sec MVICs were performed for each muscle with a 30-sec break between repetitions. To optimize maximal contractions participants were verbally motivated. The average of the three MVICs based on the RMS calculated for the intermediate three seconds of each muscle’s MVICs was considered in the normalization of the data.

### 2.6. Heart Rate and Perceived Exertion Measurements

Heart rate (HR) and perceived excursion (PE) were monitored with a sensor embedded in a 66-cm waterproof belt, which was placed on the chest at the level of the xiphoid process (Polar H10, Heart Rate Sensor, Kempele, Finland), and the Borg’s 15-point rating scale of perceived exertion was used before and at the end of each walking or running condition [55].

### 2.7. Statistical Analysis

Violations of statistical assumptions regarding the normality of the EMG data distribution, after being examined with the Shapiro–Wilk test and by visually observing the Q-Q and box plot graphs, necessitated a logarithmic transformation of the EMG raw signals. Logarithmic means and standard deviations were back-transformed and presented as geometric means and a 95% confidence interval. One-way repeated measures ANOVA was performed to detect possible differences between LLDs (0, 1, 2, and 3 cm) for stride length and time, step width as well as cadence. Differences between LLDs (within-subjects factor) and sides (LLS and SLS; between-subjects factor), as well as the LLD-by-side interaction regarding step length and time, stance, and swing phase time, GRFs, and EMG activity of the muscles under investigation were assessed using a mixed-design ANOVA. Sphericity of the data was determined based on the Mauchly’s test and significant Greenhouse–Geisser correction was used. When a statistically significant interaction between LLD and side was detected by the ANOVA test for a dependent variable, this interaction was interpreted by the visual inspection of trends presented by values and bar graphs according to how that variable was reported followed by pairwise comparisons using the Bonferroni adjustment to avoid type I statistical error. The statistical analysis of the data was performed with SPSS 28.0 (IBM Corp, Armonk, NY, USA), while the significance level was set at the level of *p* ≤ 0.05. 

## 3. Results

The methods and procedures used in the study were accepted and well tolerated by the participants. Participants were able to undergo the experimental procedure without reporting adverse effects, undue harm, or significant discomfort. 

In our datasets, <2% of data points were missing for EMG activity of the muscles under investigation only during running. The missing data appeared to be missing completely at random, as there is no discernible pattern in the missing values probably due to sensor malfunction. To address the missing data, we employed the single imputation method by using the mean value of the observed data [56].

### 3.1. Spatiotemporal Parameters of Walking and Running

Statistical analysis revealed a significant interaction between LLD and side for step length during walking (F = 36.485, *p* ≤ 0.001, partial η^2^ = 0.422). Step length significantly increased on the LLS and decreased on the SLS as LLD increased, with the LLS step length being significantly greater than the SLS step length at ≥1 cm LLDs (see Table 1 for pairwise comparisons).

A significant interaction between LLD and side was revealed for step time (F = 120.340, *p* ≤ 0.001, partial η^2^ = 0.706). Step time significantly decreased on the LLS and increased on the SLS as LLD increased (see Table 1 for pairwise comparisons). The interaction between LLD and side was also significant for the duration of the stance phase (F = 66.688, *p* ≤ 0.001, partial η^2^ = 0.572) and swing phase (F = 54.453, *p* ≤ 0.001, partial η^2^ = 0.521). The stance phase significantly increased on the LLS and decreased on the SLS as LLDs increased. Conversely, the swing phase significantly decreased on the LLS and increased on the SLS as LLDs increased (see Table 1 for pairwise comparisons).

One-way ANOVA showed a significant increase in stride length (F = 3.822, *p* ≤ 0.05, partial η^2^ = 0.133), step width (F = 10.254, *p* ≤ 0.001, partial η^2^ = 0.291), and stride time (F = 3.437, *p* ≤ 0.05, partial η^2^ = 0.121), as well as a significant decrease in cadence (F = 3.330, *p* ≤ 0.05, partial η^2^ = 0.118), as LLD increased (see Table 1 for pairwise comparisons).

Two-way ANOVA revealed a significant interaction between LLD and side (F = 3.144, *p* ≤ 0.05, partial η^2^ = 0.059) for step length during running. Step length significantly increased only on the SLS while remaining unchanged on the LLS as LLD increased (see Table 2 for pairwise comparisons).

A significant interaction between LLD and side (F = 53.382, *p* ≤ 0.001, partial η^2^ = 0.516) was revealed for step time. Step time decreased on the LLS and increased on the SLS as LLD increased (see Table 2 for pairwise comparisons). A significant interaction between LLD and side was found also for the duration of the stance phase (F = 47.903, *p* ≤ 0.001, partial η^2^ = 0.489) and swing phase (F = 26.056, *p* ≤ 0.001, partial η^2^ = 0.343). The duration of the stance phase significantly increased on the LLS and decreased on the SLS, while the duration of the swing phase significantly decreased on the LLS and increased on the SLS as LLD increased (see Table 2 significantly for pairwise comparisons).

Stride length (F= 2.799, *p* ≤ 0.05, partial η^2^ = 0.101) and stride time (F= 2.867, *p* ≤ 0.05, partial η^2^ = 0.103) as well as step width (F= 5.814, *p* ≤ 0.001, partial η^2^ = 0.189) significantly increased as LLD increased. Cadence was not significantly decreased during running (see Table 2 for significant pairwise comparisons).

### 3.2. Ground Reaction Forces Exerted during Walking and Running

Two-way ANOVA revealed significant interactions between LLD and side for GRFs exerted on the rearfoot (F = 15.293, *p* < 0.001, partial η^2^ = 0.234), midfoot (F = 24.501, *p* < 0.001, partial η^2^ = 0.329), and forefoot (F = 18.583, *p* < 0.001, partial η^2^ = 0.271) during walking. Pairwise comparisons between LLDs and sides for ground reaction forces exerted in all plantar areas during walking are depicted in Figure 3.

GRFs on the rearfoot of the LLS remained almost unchanged, while GRFs on the rearfoot of the SLS increased as LLD increased. Both the LLS and SLS midfoot GRFs were increased as LLD increased, with GRFs on the LLS increasing more compared to the SLS. GRFs on the forefoot of the LLS decreased but remained unchanged on the forefoot of the SLS as LLD increased, with GRF on the LLS being lower compared to the SLS in all LLDs. 

Ground reaction forces exerted on the rearfoot of both the LLS and SLS were not significantly affected by LLD during running. A significant interaction between LLD and side was found for midfoot GRFs (F = 10.933, *p* < 0.001, partial η^2^ = 0.179). Midfoot GRFs of both sides increased as LLD increased, with a greater increase seen in the LLS compared to SLS. Two-way ANOVA also revealed a significant interaction between LLD and side for forefoot GRFs (F = 15.549, *p* < 0.001, partial η^2^ = 0.237). GRFs decreased in the forefoot of the LLS while remaining unchanged in the forefoot of the SLS as LLD increased. Pairwise comparisons between LLDs and sides for ground reaction forces exerted in all plantar areas are depicted in Figure 4.

### 3.3. EMG Activity of Hip and Trunk Muscles during Walking and Running

Statistical analysis showed nonsignificant main effects of LLD and side as well as a nonsignificant interaction between LLD and side for the EMG activity of the thoracic and lumbar erector spinae and gluteus medius during both walking and running (Figure 5 and Figure 6). 

Gluteus medius EMG activity was greater compared to thoracic and lumbar erector spinae activity during both walking and running, and all muscles demonstrated greater EMG activity during running compared to walking (identification of the aforementioned differences was based on visual inspection rather than applying statistical tests to the data as such analysis would be beyond the scope of this study).

Absolute differences between sides for EMG activity ranged from 0.3% to 1.8% for TES, 0.3% to 1.0% for LES, and 0.4% to 3.8% for GLU regardless of LLD and speed tested.

### 3.4. Heart Rate and Perceived Exertion

Significant main effects of LLD (F = 3.180, *p* ≤ 0.05, partial η^2^ = 0.113) and phase (F = 160.308, *p* < 0.001, partial η^2^ = 0.865) were found for HR, while main effects of phase only were significant (F = 16.286, *p* ≤ 0.001, partial η^2^ = 0.394) for PE during walking. Main effects of phase were also significant for HR (F = 453.652, *p* ≤ 0.001, partial η^2^ = 0.948) and PE (F = 51.755, *p* ≤ 0.001, partial η^2^ = 0.674). Pairwise comparisons revealed significant differences between the start and end of the walking and running trial in each testing condition for both HR and PE; however, the differences between testing conditions were not significant either for HR or PE (see Table 3 for pairwise comparisons).

## 4. Discussion

### 4.1. Effects of LLD on Spatiotemporal Gait Parameters

The results of the present study showed that a 1–3 cm artificially induced LLD significantly affected the symmetry of walking and running in terms of spatiotemporal and GRFs of the gait cycle without bilaterally affecting the EMG activity of the gluteus medius and erector spinae.

Regarding spatial parameters, step length decreased in the SLS and increased in the LLS as LLD increased during walking as opposed to the increase in the SLS step length and an almost unchanged step length in the LLS that occurred during running. Furthermore, changes in step length during walking were evident (in LLS) with ≥1 cm LLD but it only became significant during running (in SLS) when LLD was 3 cm. These changes significantly increased stride length during both walking and running; however, this change was no more than 1% at 3 cm LLD compared to stride length without LLD. Furthermore, step width was significantly increased as LLD increased, being more pronounced at 3 cm LLD during walking and ≥2 cm LLD during running compared to step width with less LLDs. Temporal parameters demonstrated significant changes during both walking and running with ≥1 cm LLD. Step time increased in the SLS and decreased in the LLS as LLD increased. Furthermore, the duration of the gait cycle increased with the LLS in stance and the SLS in swing phase as the duration of both phases increased. In contrast, the duration of a consecutive gait cycle in which the SLS was in the stance and the LLS was in the swing phase was decreased as the duration of these gait phases decreased. These variations in the duration of consecutive gait cycles contributed to no more than a 1.0% increase in stride time and a 1.0% decrease in cadence during locomotion at 3 cm LLD compared to no-LLD (Figure 7).

The decrease in SLS step length compared to LLS step length as well, as the minimal albeit significant changes in stride length found in the present study during treadmill walking, were supported by previously recorded data on an instrumented treadmill [57] as well as during overground walking in subjects with either artificially induced LLD [58,59] or anatomical LLD of ≥2 cm [40]. In accordance with our findings, step width has also been found to be increased as LLD increased, with some authors detecting significant differences in subjects with artificially induced LLD of 2 cm during treadmill walking [57] but not with 3 cm LLD during overground walking [59]. In addition, previous studies performed in subjects with artificially induced LLD up to 3 cm during treadmill walking confirmed the increased SLS step time compared to LLS step time, and the minor changes in stride time among LLDs [57] found in the present study. Furthermore, our findings were similar to those of a previous study recorded during overground walking regarding increased LLS stance phase time but not decreased SLS stance phase time as LLD increased [17]. Others, however, have shown a decrease in the stance phase time and an increase in swing phase time but for both feet [24].

The asymmetrical gait, as manifested by the altered spatiotemporal parameters of walking and running, could be attributed to the compensatory mechanisms that subjects with LLD often exhibit to maintain symmetry and minimize the vertical shift of the body’s center of mass. Khamis and Carmeli’s [33] systematic review of findings on the effects of LLD on lower extremity kinematics concluded that gait symmetry in these individuals is maintained by compensatory mechanisms that require deviations of the lower limb joints and pelvis at both the sagittal and frontal planes, which are increased as the discrepancy increases. When the SLS is in the stance phase, individuals with LLD tend to functionally lengthen their leg by increasing foot supination and ankle plantar flexion and decreasing the flexion at the knee and hip joints. These responses, combined with the kinematic deviations exhibited by the LLS as it swings through the same gait cycle (increased ankle dorsiflexion, knee, and hip flexion) [33] may partially account for the step, stride, and the corresponding gait cycle spatial and temporal changes observed. The increased foot pronation, ankle dorsiflexion, as well as knee and hip flexion of the LLS as it tends to shorten when in stance phase, combined with decreased ankle dorsiflexion and hip abduction of the SLS as it swings, may account for the spatial and temporal changes that occur during a successive gait cycle [33]. 

The decrease in SLS step length that occurred during walking may be justified by the compensatory responses required to maintain symmetrical support and balance during the double-stance phase, a situation not encountered during running. Conversely, the increased SLS step length in running may be due to the unimpeded movement of both legs during the flight phase, as both legs were simultaneously off the ground. The increased hip abduction expected to be achieved by the SLS either being in the stance or swing phase of gait and the increased hip adduction and abduction achieved by the LLS in the stance and swing phase, respectively, may account for the wider step induced during both walking and running. The increase in step width found with increased LLD may be one of the factors that contributed to the small changes achieved in stride length and cadence during locomotion as the increase in step length of one side was not compensated by an equal decrease in step length of the opposite side (Figure 7). The increased stride width can also help individuals achieve a wider base of support, allowing better control during walking and running thereby reducing the risk of falling or tripping.

### 4.2. Effects of LLD on GRFs

The significant interaction found between LLD and side for GRFs revealed that the rearfoot, midfoot, and forefoot areas in the LLS and SLS responded differently to LLD during both walking and running. GRFs in the rearfoot of the LLS remained unchanged but increased in the SLS, increased in the midfoot of both sides but more so in the LLS, and decreased on the forefoot of the LLS while remaining unchanged in the SLS. In general, GRFs were lower in the rearfoot and forefoot and greater in the midfoot of the LLS compared to the SLS, with some differences being evident at 1.0 cm LLD. Overall GRFs were greater on the SLS compared to LLS.

Our findings agree with the findings presented in the literature based on which LLD of 0.5–4.0 cm creates an asymmetry in vertical GRFs and/or plantar pressure during overground walking mostly at the preferred speed by being greater on the short leg compared to the longer [16,17,18,20,27,28]. This asymmetry is more noticeable at >1.5 cm LLD [20] and is not affected by load carriage [18], phase of walking (e.g., single-leg or double-leg stance [20]), or type of LLD (anatomic or artificially induced [16]). In contrast to our findings, other authors have reported greater GRFs or plantar pressures in the longer compared to the shorter limb [19,29,30] and differences in the distribution of forces between plantar foot areas (e.g., increased forefoot and unchanged hindfoot SLS loading [28]), probably due to the different methods implemented in the investigation of LLD.

The distribution of GRFs both within plantar areas of the same foot and between the LLS and SLS foot for the same plantar area could be explained by alterations in the loading patterns between limbs and/or the compensatory responses expected to occur in lower limb kinematics to offset the LLD-related biomechanical challenges required for walking or running. Swaminathan et al., in a previous study, showed that the distribution of body mass increased in the shorter leg when a 3.5 cm LLD was simulated in the upright standing position, probably due to a shift of the center of mass towards the side of the shorter limb [60]. These findings partially support our results as overall GRFs were greater in the SLS than in the LLS. Alterations in lower extremity kinematics could also explain the distribution of GRFs between the plantar areas of each foot. The greater GRFs exerted on the rearfoot of the SLS may result from the greater impact generated by the short leg, as it is required to be transferred from a greater vertical height during the transition from the long leg to the short leg in the stance phase of walking [20]. Knee joint restriction, which is expected to be encountered in the gait cycle of individuals with LLD where the longer leg tends to shorten during the stance phase, may eventually increase the ground reaction forces exerted, especially on the midfoot area of the LLS. Previous studies indirectly confirm our observations as they have shown that similar artificially induced restrictions in the knee joint flexion (or extension) increase peak ground reaction forces during walking [61,62]. These changes have been observed in both the limb with restricted and the contralateral limb with unrestricted movement of the knee joint at distinct phases of gait, although significant differences were found only for initial contact [61]. The fact that the GRFs exerted on midfoot areas of both feet were progressively increased as LLD increased may also be related to the greater pronation and supination expected to be presented by the LLS and SLS foot, respectively, during walking [33], as has been shown in individuals with asymptomatic pronated and supinated feet [63]. The progressive reduction in GRFs exerted on the forefoot of the LLS could be explained by the reduced dorsi flexion that usually occurs by this leg when it is in the stance phase of the gait cycle in LLD patients [33]. These findings agree with those reported in a previous study where GRFs were reduced immediately after restricting ankle dorsiflexion with a custom-made ankle brace in healthy individuals [64]. Finally, the increased ankle plantar flexion and abduction typically experienced by the SLS as it pushes off may account for the greater GRFs seen in this limb compared to the contralateral LLS limb [33].

The reaction forces exerted on the foot when it contacts the ground, such as when walking or running, have also a direct effect on the contact forces exerted on the ankle, knee, and hip joints, as well as on the pelvis. In this context our findings suggest that the increased GRFs exerted on the rearfoot, midfoot, and forefoot plantar area of the SLS will increase the contact forces of the overlying joints accordingly as LLD increased. Indeed, previous studies have shown that the cumulative contact forces on the joints that comprise the kinetic chain of the lower limb are increased as LLD increases [16,65]. Relatively lower GRFs exerted on the plantar regions of the LLS foot compared to the SLS foot may also indicate lower contact forces at the overlying joints of the ipsilateral limb, a finding also verified by previous studies [16,65]. This information could be used by clinicians to evaluate whether GRFs are evenly distributed by implementing therapeutic means for the management of LLD, thus preventing excessive contact forces on the lower limb joints and possible onset of osteoarthritis.

### 4.3. Effects of LLD on EMG of Trunk and Hip Musculature

The current findings regarding EMG activity of pelvic and trunk controllers in the frontal plane, namely the gluteus medius and the thoracic and lumbar erector spinae, agree with most findings reported in the literature. Vink and Huson [24] did not find significant changes in the EMG activity of intrinsic lumbar muscles during overground walking with artificially induced LLD of 4.0 cm. Gurney et al. failed also to demonstrate an increase in gluteus medius EMG activity by artificially increasing LLD up to 4 cm in healthy individuals between 55 and 86 years of age [31]. Bird et al. [32] showed nonsignificant changes in the amplitude of the EMG activity of gluteus medius and erector spinae at the third lumbar vertebra (L3) during walking by applying unilaterally a heel lift of up to 2 cm height compared to a barefoot condition. Other authors have reported that EMG activity of the erector spinae at the third lumbar vertebrae did not demonstrate a significant increase during standing but only when LLD was increased artificially by more than 3.5 cm [66].

The complex interactions between the joints consisting of the kinematic chains of the lower limbs expected to occur in the sagittal and frontal plane to compensate for LLD may eventually prevent excessive pelvis obliquity and lumbar scoliosis during walking and running, limiting the changes in the EMG activity of the muscles under investigation [67]. Although pelvis and lumbar spine may deviate by approximately 10° in the frontal plane during walking, as has been shown in adolescent patients with structural LLD of 2.0–6.0 cm [68], such deviations are not so pronounced in healthy individuals who have been subjected to artificially induced LLD. A previous study showed that pelvic obliquity may compensate for mild LLD up to 2.0 cm [69], but it is not expected to be pronounced. Using dynamic rasterstereographic analysis, some authors showed that artificially induced LLD up to 2.0 cm can induce pelvic obliquity and torsion that does not exceed 1.0 cm and 2.0°, respectively, with minor changes only in trunk lateral deviation and torsion [11,12]. Others showed that the mean pelvic rotation angle in the frontal plane was changed by 1.52 degrees when walking with an artificially induced LLD of 4.0 cm [24]. Moreover, Kakushima et al. [70] revealed that walking with an orthotic device that raised the heel by 3 cm increased maximum lateral bending of the lumbar spine by 3° (from 6.1° to 9.1°) and thoracic spine by 1.2° (from 3.0° to 4.2°). Even if the deviations that occurred in the lumbopelvic complex were considered significant, the EMG activity of the adjacent muscles may not have ultimately needed to increase as the compensated motions of the adjacent joints “absorbed” the induced postural changes. For example, the torsion of the sacroiliac joints expected to occur on the LLS [1,11,13] may prevent excessive vertebral asymmetry. In addition, the asymmetric change in the height of the intervertebral discs and the increase in loading of the zygapophyseal joints that occur in single functional units of the spine, such as those occurring in LLD patients with functional scoliosis [5], may be small, limiting the need for compensative muscular responses. Apparently, these alterations have failed to activate the joint receptors at the critical point required to cause muscle contractions of the paraspinal and hip muscles. Previous investigators have shown in animals (pigs) that such reflex muscle responses can be elicited in the quadratus lumborum following stimulation of the ventral region of the sacroiliac joint capsule [71]. They have also been induced in paraspinal muscles for the correction of balance, following activation of joint receptors in the lumbar spine that are stimulated by trunk rotation [72]. 

### 4.4. Clinical Implications

The asymmetric exertion of GRFs on the plantar surfaces of the foot during locomotion activities such as this demonstrated in the present study may provide valuable information about the areas of the foot that are at higher risk of injury. Discrepancies in leg length have been identified as a contributing factor in numerous sports-related injuries. These injuries encompass stress fractures occurring in the metatarsals, tibia, and femur. Such fractures have been observed to occur in the shorter [73], the longer [74], or both the shorter and longer limb [73]. Other researchers have shown that the longer limb is more susceptible to foot pain and injuries such as plantar fasciitis [7]. In contrast, clinicians should search for other possible sources of lumbopelvic pain in individuals with LLD as, based on the present findings, it is highly unlikely to be caused by increased EMG activity of the pelvic and trunk muscles. 

Moreover, the identification of musculoskeletal abnormalities is a major concern of clinical therapists seeking to avoid adverse effects on a person’s functional ability and prevent injuries. Our findings revealed that while bilateral differences in terms of spatiotemporal gait parameters and the associated GRFs could be detected in individuals with LLD as low as 1 cm during walking, most of the changes during running were evident only when LLD was ≥2 cm. In this context, treadmill-based gait analysis proposed in the present study can be a useful tool in the assessment of LLD as its presence is often overlooked [3]. It may also allow for a more individualized approach to intervention, enabling clinicians to assess how different therapeutic treatments, such as shoe lifts of varying heights, affect the patient’s locomotion under standardized conditions (e.g., speed, duration, and parameters of gait analysis). Furthermore, periodic assessment of gait parameters using an instrumented treadmill may enable monitoring treatment efficacy by objectively measuring changes in gait mechanics, determining whether the chosen intervention is producing the desired outcomes and adjusting the treatment plan if necessary.

### 4.5. Study Limitations

Certain limitations related to the study sample and the implemented methodology may prevent generalization of our findings to the population at large. The sample consisted of both males and females who present differences in pelvis and trunk structure as well as muscle geometry [75,76,77], gait kinematics, spatiotemporal parameters, and plantar pressure distribution [78,79,80]. Hence, direct comparisons with results presented from studies including only males or females and/or the general population should be performed with caution. Additionally, the artificially induced LLD in the present study was more of a simulation of sudden-onset LLD, such as that resulting from surgery (e.g., hip replacement) [81] or trauma (e.g., fracture) [8] rather than a long-term structural deviation similar to that experienced by a child with LLD, which could present different outcomes [16,82,83]. Our findings should also be viewed in the context of the limited time each participant was given for treadmill-based walking and running familiarization in each LLD. More time for familiarization with the study conditions, having caused an asymmetry to which the participants were not accustomed, could have affected the results. 

Furthermore, LLD was simulated by adding layers of rubber in an orthotic shoe lift designed for compensating temporary LLD induced by wearing an orthotic walking boot or a postoperative shoe. Although shoe lifts were placed over running shoes on both feet to increase their size equally and avoid possible effects on the spatiotemporal parameters of gait, the additional rubber layers used to increase height to one limb may have attenuated impact forces, masking the true effect of LLD itself [84,85]. Finally, our findings are also limited to the 1–3 m LLD generated in the present study; a larger LLD could further enhance the walking and running asymmetry. 

## 5. Conclusions

The findings of this study revealed that walking and running symmetry was significantly affected by experimentally induced LLD of 1–3 cm in terms of spatiotemporal characteristics and GRFs with some bilateral differences being evident at 1 cm LLD. Step length decreased on the SLS and increased on the LLS as LLD increased during walking. In contrast, step length increased on the SLS but remained almost unchanged on the LLS as LLD increased during running. Although there was an increase in stride length at both speeds, it was not more than 1% of the stride length achieved without LLD. The increase in LLD during both walking and running caused a significant increase in step width. 

Step time decreased in the LLS and increased in the SLS as LLD increased during both walking and running, significantly changing stride time but not by more than 1% compared to non-LLD. Furthermore, stance time increased in the LLS and decreased in the SLS during both walking and running, while the opposite occurred for swing time. Cadence decreased as LLD increased although only significantly during walking. GRFs in the rearfoot of the LLS remained unchanged but increased in the SLS, increased in the midfoot of both sides but more so in the LLS, and decreased in the forefoot of the LLS, while remaining unchanged in the SLS as LLD increased. LLD had no impact on the bilateral EMG activity of thoracic and lumbar trunk extensors and hip abductors. Despite limitations related to the sample and methodology used in this study, instrumented walking and/or treadmill running could be a useful means of identifying, as well as monitoring and controlling, the effectiveness of interventions used to manage LLD.

## Figures and Tables

**Figure 1 sensors-23-09695-f001:**
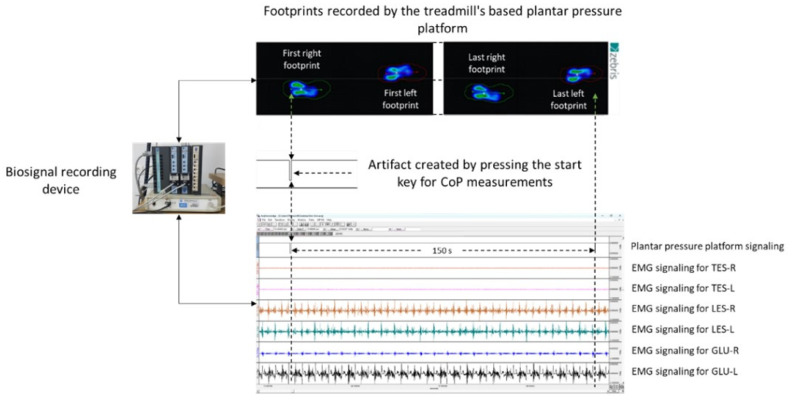
Graphical representation of the method used to determine the time frame of right (R) and left (L) thoracic and lumbar erector spinae (TES and LES) and gluteus medius (GLU) EMG activity based on the number of footprints generated by center of pressure (CoP) recordings during treadmill walking and running.

**Figure 2 sensors-23-09695-f002:**
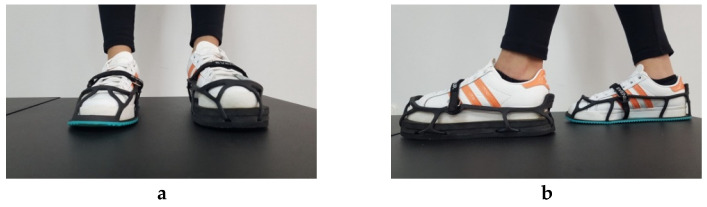
Front (**a**) and side (**b**) view of the shoes (EVEN Up, OPED, Oberlaindern, Germany) used to induce artificial leg length discrepancy by 2 cm.

**Figure 3 sensors-23-09695-f003:**
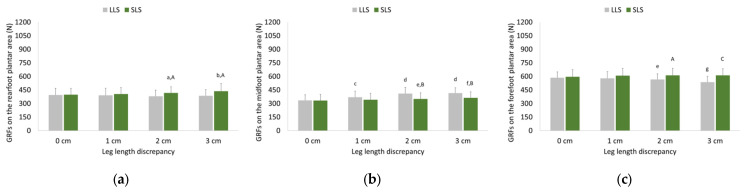
Means and standard deviations (error bars) of ground reaction forces exerted on (**a**) the rearfoot, (**b**) the midfoot, and (**c**) the forefoot plantar areas during walking without (0 cm) and with artificially induced leg length discrepancy of 1, 2, and 3 cm. ^a^ Significant different (SD) compared to 0 cm LLD (*p* ≤ 0.01); ^b^ SD compared to 0 and 1 cm (*p* ≤ 0.001) as well as 2 cm LLD (*p* ≤ 0.01); ^c^ SD compared to 0 cm LLD (*p* ≤ 0.001); ^d^ SD compared to 0 cm and 1 cm LLD (*p* ≤ 0.001); ^e^ SD compared to 0 cm LLD (*p* ≤ 0.05); ^f^ SD compared to 0 cm (*p* ≤ 0.001) and 1 cm LLD (*p* ≤ 0.01); ^g^ SD compared to 0, 1 and 2 cm LLD (*p* ≤ 0.001); ^A^ SD compared to LLS (*p* ≤ 0.05); ^B^ SD compared to LLS (*p* ≤ 0.01); ^C^ SD compared to LLS (*p* ≤ 0.001); GRFs: ground reaction forces; LLS: long-limb side; SLS: short-limb side.

**Figure 4 sensors-23-09695-f004:**
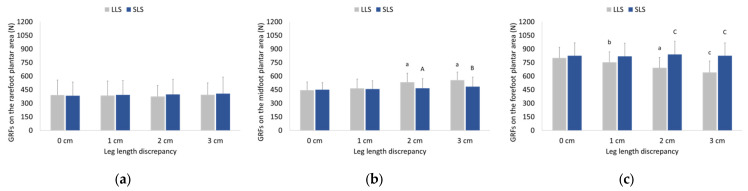
Means and standard deviations (error bars) of ground reaction forces exerted on (**a**) the rearfoot, (**b**) the midfoot, and (**c**) the forefoot plantar area during walking without (0 cm) and with artificially induced leg length discrepancy of 1, 2, and 3 cm. ^a^ Significant different (SD) compared to 0 and 1 cm LLD (*p* ≤ 0.001); ^b^ SD compared to 0 cm LLD (*p* ≤ 0.05); ^c^ SD compared to 0 and 1 cm (*p* ≤ 0.001), as well as 2 cm LLD (*p* ≤ 0.01); ^A^ SD compared to LLS (*p* ≤ 0.05); ^B^ SD compared to LLS (*p* ≤ 0.01); ^C^ SD compared to LLS (*p* ≤ 0.001); GRFs: ground reaction forces; LLS: long-limb side; SLS: short-limb side.

**Figure 5 sensors-23-09695-f005:**
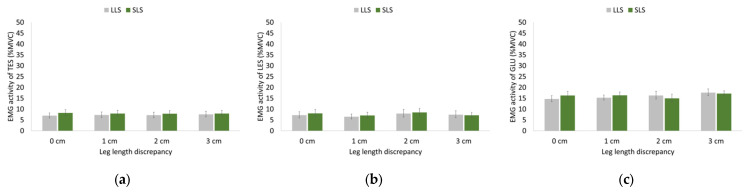
Geometric means and 95% confidence intervals (error bars) of (**a**) thoracic erector spinae (TES), (**b**) lumbar erector spinae (LES), and (**c**) gluteus medius (GLU) EMG activity on the long-limb side (LLS) and short-limb side (SLS) during walking without (0 cm) and with artificially induced leg length discrepancy of 1, 2, and 3 cm.

**Figure 6 sensors-23-09695-f006:**
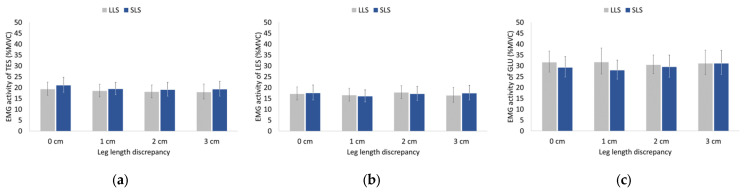
Geometric means and 95% confidence intervals (error bars) of (**a**) thoracic erector spinae (TES), (**b**) lumbar erector spinae (LES), and (**c**) gluteus medius (GLU) EMG activity on the long-limb side (LLS) and short-limb side (SLS) during running without (0 cm) and with artificially induced leg length discrepancy of 1, 2, and 3 cm.

**Figure 7 sensors-23-09695-f007:**
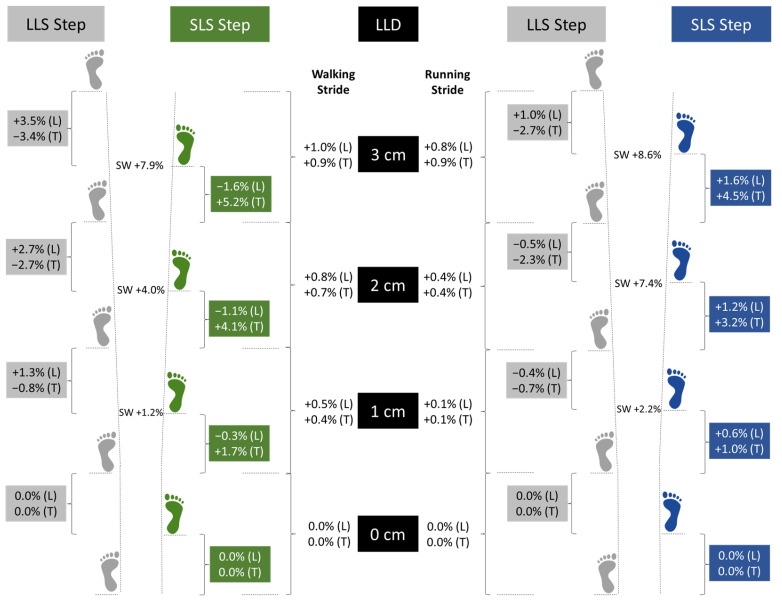
Graphical representation of changes on the long-limb side (LLS) and short-limb side (SLS) step and stride length (L) and time (T) as well as step width (SW) during walking (green colored side) and running (blue colored side) with artificially induced leg length discrepancy (LLD) of 1, 2, and 3 cm, calculated as percent increase (+) or decrease (−) of values recorded during walking without LLD (0 cm).

**Table 1 sensors-23-09695-t001:** Mean, standard deviations, and percentage change in spatiotemporal parameters of gait during walking without (0 cm) and with artificially induced leg length discrepancy of 1, 2, and 3 cm.

Gait Parameters	Side	Leg Length Discrepancy			
		0 cm	1 cm	2 cm	3 cm
Step length (cm)	LLS	73.28 ± 2.84	74.21 ± 2.58 ^a^	75.23 ± 2.71 ^b,A^	75.85 ± 3.36 ^b,A^
	SLS	73.27 ± 2.59	73.04 ± 2.41	72.46 ± 2.22 ^c^	72.12 ± 2.47 ^d^
Stride length (cm)		146.55 ± 5.20	147.25 ± 4.72	147.69 ± 4.52	147.98 ± 5.37
Step width (cm)		9.67 ± 2.38	9.79 ± 2.53	10.06 ± 2.63	10.44 ± 2.54 ^e^
Step time (s)	LLS	0.53 ± 0.02	0.52 ± 0.02 ^c^	0.51 ± 0.02 ^b,A^	0.51 ± 0.02 ^b,A^
	SLS	0.52 ± 0.02	0.53 ± 0.02 ^a^	0.55 ± 0.02 ^b^	0.55 ± 0.02 ^f^
Stride time (s)		1.05 ± 0.04	1.06 ± 0.03	1.06 ± 0.03	1.06 ± 0.04
Stance phase time (s)	LLS	0.65 ± 0.02	0.66 ± 0.02 ^a^	0.67 ± 0.02 ^b,A^	0.68 ± 0.02 ^b,A^
	SLS	0.66 ± 0.03	0.65 ± 0.02	0.65 ± 0.02 ^b^	0.65 ± 0.03 ^b^
Swing phase time (s)	LLS	0.40 ± 0.02	0.39 ± 0.02 ^c^	0.39 ± 0.02 ^e,A^	0.39 ± 0.02 ^e,A^
	SLS	0.40 ± 0.02	0.40 ± 0.01 ^a^	0.41 ± 0.01 ^b^	0.42 ± 0.02 ^f^
Cadence (step·min^−1^)		114.16 ± 4.09	113.66 ± 3.66	113.30 ± 3.50	113.15 ± 4.14

^a^ Significant different (SD) compared to 0 cm LLD (*p* ≤ 0.001); ^b^ SD compared to 0 and 1 cm LLD (*p* ≤ 0.001); ^c^ SD compared to 0 cm LLD (*p* ≤ 0.05); ^d^ SD compared to 0 cm (*p* = 0.01) and 1 cm LLD (*p* ≤ 0.05); ^e^ SD compared to 0 cm (*p* ≤ 0.001) and 1 cm LLD (*p* ≤ 0.01); ^f^ SD compared to 0 and 1 cm LLD (*p* ≤ 0.001), as well as 2 cm LLD (*p* ≤ 0.05); ^A^ SD compared to SLS (*p* ≤ 0.001); LLS: long-limb side; SLS: short-limb side.

**Table 2 sensors-23-09695-t002:** Mean and standard deviations of spatiotemporal parameters of gait during running without (0 cm) and with artificially induced leg length discrepancy of 1, 2, and 3 cm.

Gait Parameters	Side	Leg Length Discrepancy			
		0 cm	1 cm	2 cm	3 cm
Step length (cm)	LLS	83.09 ± 4.10	82.77 ± 3.87	82.72 ± 3.60	83.14 ± 3.99
	SLS	83.23 ± 3.83	83.71 ± 4.03	84.24 ± 3.99	84.57 ± 4.00 ^a^
Stride length (cm)		166.32 ± 7.75	166.50 ± 7.68	166.96 ± 7.21	167.72 ± 7.46
Step width (cm)		6.84 ± 3.09	6.99 ± 3.14	7.34 ± 3.03 ^b^	7.43 ± 2.96 ^c^
Step time (s)	LLS	0.37 ± 0.02	0.37 ± 0.02	0.37 ± 0.02 ^d,A^	0.36 ± 0.02 ^e,A^
	SLS	0.37 ± 0.02	0.38 ± 0.02	0.38 ± 0.02 ^e^	0.39 ± 0.02 ^f^
Stride time (s)		0.75 ± 0.03	0.75 ± 0.03	0.75 ± 0.03	0.75 ± 0.03
Stance phase time (s)	LLS	0.32 ± 0.03	0.33 ± 0.03 ^g,B^	0.34 ± 0.02 ^e,A^	0.35 ± 0.02 ^f,A^
	SLS	0.32 ± 0.03	0.32 ± 0.02	0.31 ± 0.02 ^h^	0.31 ± 0.02 ^b^
Swing phase time (s)	LLS	0.43 ± 0.03	0.42 ± 0.04 ^b^	0.41 ± 0.03 ^i,A^	0.41 ± 0.03 ^d,A^
	SLS	0.43 ± 0.03	0.43 ± 0.04	0.44 ± 0.03 ^j^	0.44 ± 0.04 ^d^
Cadence (step·min^−1^)		161.11 ± 7.78	161.20 ± 8.37	160.40 ± 7.08	159.81 ± 7.48

^a^ Significant different (SD) compared to 0 cm LLD (*p* ≤ 0.01); ^b^ SD compared to 0 cm LLD (*p* ≤ 0.05); ^c^ SD (marginally) compared to 0 cm (*p* = 0.054) and 1 cm LLD (*p* ≤ 0.05); ^d^ SD compared to 0 cm (*p* ≤ 0.001) and 1 cm LLD (*p* ≤ 0.01); ^e^ SD compared to 0 cm and 1 cm LLD (*p* ≤ 0.001); ^f^ SD compared to 0 cm and 1 cm LLD (*p* ≤ 0.001) as well as 2 cm LLD (*p* ≤ 0.05); ^g^ SD compared to 0 cm LLD (*p* ≤ 0.001); ^h^ SD compared to 0 and 1 cm LLD (*p* ≤ 0.01); ^i^ SD compared to 0 (*p* ≤ 0.001) and 1 cm LLD (*p* ≤ 0.05); ^j^ SD compared to 0 (*p* ≤ 0.01) and 1 cm LLD (*p* ≤ 0.05); ^A^ SD compared to SLS (*p* ≤ 0.001); ^B^ SD compared to SLS (*p* ≤ 0.05); LLS: long-limb side; SLS: short-limb side.

**Table 3 sensors-23-09695-t003:** Means ± standard deviations of heart rate and perceived exertion during walking and running without (0 cm) and with artificially induced leg length discrepancy of 1, 2, and 3 cm.

Parameters	Phase	Leg Length Discrepancy			
		0 cm	1 cm	2 cm	3 cm
Walking					
HR (beats·min^−1^)	Start	82.0 ± 10.0 ^A^	84.5 ± 12.9 ^A^	84.3 ± 12.2 ^A^	85.3 ± 12.3 ^A^
	End	99.1 ± 13.7	100.2 ± 13.8	100.6 ± 13.9	102.9 ± 12.4
PE (score)	Start	7.4 ± 1.3 ^B^	7.2 ± 1.1 ^C^	7.1 ± 1.2 ^A^	6.8 ± 1.0 ^a,A^
	End	7.8 ± 1.7	7.7 ± 1.6	7.7 ± 1.4	7.7 ± 1.6
Running					
HR (beats·min^−1^)	Start	88.4 ± 11.8 ^A^	87.6 ± 11.7 ^A^	89.9 ± 12.0 ^A^	90.0 ± 8.3 ^A^
	End	149.9 ± 17.9	148.6 ± 18.4	147.8 ± 22.9	149.4 ± 21.7
PE (score)	Start	8.1 ± 1.5 ^A^	7.8 ± 1.5 ^A^	7.5 ± 1.9 ^A^	8.2 ± 1.8 ^A^
	End	10.2 ± 2.6	10.4 ± 2.7	10.1 ± 2.6	10.7 ± 2.7

^a^ Significant different (SD) compared to 0 and 1 cm LLD (*p* ≤ 0.05); ^A^ SD compared to end (*p* ≤ 0.001); ^B^ SD compared to end (*p* ≤ 0.05); ^C^ SD compared to end (*p* ≤ 0.01); HR: heart rate; PE: perceived exertion.

## Data Availability

The data presented in this study are available upon request from the corresponding author.

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
