# Peer review of "Effects of Artificially Induced Leg Length Discrepancy on Treadmill-Based Walking and Running Symmetry in Healthy College Students: A Lab-Based Experimental Study"

_sensors, 2023, doi:10.3390/s23249695_

Round 1

Reviewer 1 Report

Comments and Suggestions for Authors

This study examines effects of LLD on gait symmetry by spatiotemporal parameters and GRF, EMG activity of muscles acting on the hip and lumbopelvic region during walking and running. This study has a lot going for it. It thoroughly collected a wide range of data, examining gait parameters, ground reaction forces, and muscle activity in the hip and lower back area. This comprehensive approach allowed for a deep analysis of how LLD affects various aspects of walking and running. However, there are some issues and limitations in this study. It didn't thoroughly discuss relevant existing literature, had major hiccups in the research design and methodology, and the conclusions weren't as clear and concise as they could be. Some sentences and expressions in the study need to be rephrased for better readability and flow. Overall, the conclusions could benefit from being more specific and explicit to highlight the significant findings and contributions of the research.

Line 42-43: "....low as 1.0 cm may need to be corrected as it can cause complications in the human body in the long term," a literature support is need.

Line 59-61: “The induced muscle imbalances are thought to potentially affect body symmetry and shock absorbing capabilities in the joints and the surrounding tissues leading to lower limb injuries and/or to pain and discomfort in the lumbar region.” a literature support is need.

Line 78-80: “Moreover, …..to control the therapeutic interventions used to treat it” Is there any relevant clinical case or actual research supporting this statement?

Line 108-109: Are there any tests of foot type such as high/low arch?  

Line 134-135: Is there any rationale or relevant research supporting the choice of test speeds (5.6 and 8.1 km/hr) for walking and running in this study?

Line 134-135: The subjects wearing shoes before putting on the special rubber shoe (as shown in Figure 2). Is there a standardized criterion for the type of shoe used?

Line 149: Please specify HR.

Line 213-217: Please provide a specific description of which variables were tested by single-factor ANOVA and two-factor ANOVA. In the context of the content describing the comparison of LLD (in four conditions) and lower limbs (in two conditions), a two-factor ANOVA should suffice. Furthermore, if a two-factor ANOVA yields an interaction effect, which indicates that the two factors are not independent and their combined effect differs from what would be expected if they were independent, how should this interaction be handled?

Line 222-358: The overall results section appears rather complex, with significant differences marked, including p-values < .05, .01, and .001. It is essential to understand the statistical significance of p-values and only distinguish between "significant" and "non-significant" differences between variables. Overly detailed categorization labeling does not contribute to the interpretation of the results.

Line 365-381: Are there any relevant research literature that can be compared with the results of this study?

Line 434-478: This passage primarily describes the results but does not provide sufficient discussion or explanations to help readers understand the significance of these findings and the underlying mechanisms.

Line 512: what does the “longer limb(s)” mean?

Line 479-516: The entire passage includes numerous comparisons with past research but lacks in-depth discussion.

Line 518-520: The sentences may need to be restructured.

Line 542-564: The lack of a comparison of gender differences is a significant research limitation. Has this study employed experimental controls or statistical methods to reduce the potential result bias that might arise from gender differences?

Line 565-579: The conclusion lacks clarity and comprehensiveness and requires a clearer summary and conclusion.

Author Response

Dear Editor, Dear reviewer

Thank you for the effort you put into revising the manuscript thus giving us the opportunity to optimize its content. All reviewer’s comments and questions were answered by the corresponding author and are as follows:

  1. Reviewer’s comment on Line 42-43: ".... low as 1.0 cm may need to be corrected as it can cause complications in the human body in the long term," a literature support is need.

Authors’ response: Citations related to what is mentioned in the specific part of the manuscript have been added according to the reviewer's comment. It has been reported that LLD as low as 1 cm is associated with low back pain, plantar fasciitis and knee osteoarthritis (Lines 44-46).

  1. Reviewer’s comment on Line 59-61: “The induced muscle imbalances are thought to potentially affect body symmetry and shock absorbing capabilities in the joints and the surrounding tissues leading to lower limb injuries and/or to pain and discomfort in the lumbar region.” a literature support is need.

Authors’ response: The sentence was modified, and citations were added to justify the quoted information (Lines 63-66).

  1. Reviewer’s comment on Line 78-80: “Moreover ….to control the therapeutic interventions used to treat it” Is there any relevant clinical case or actual research supporting this statement?

Authors’ response: Controlling the effect of LLD corrective means such as orthotics has so far been studied kinematically in individuals with structural LLD using motion capture systems. The ability to identify LLD using an instrumented treadmill may allow similar control on the effect of corrective means. Certain modifications to the manuscript were made and relevant references were used to comply with the reviewer's instructions (Lines 84-86).

  1. Reviewer’s comment on Line 108-109: Are there any tests of foot type such as high/low arch?

Authors’ response: Foot type in the present study was determined by the widely used clinical test of FPI-6, a six-point scale (one point of which is arch height), based on which the foot is classified as normal, pronated, overpronated, or supinated (Lines 120-125).

  1. Reviewer’s comment on Line 134-135: Is there any rationale or relevant research supporting the choice of test speeds (5.6 and 8.1 km/h) for walking and running in this study?

Authors’ response: Both speeds were chosen based on their applicability to physical activities by healthy active young individuals. The speed of 5.6 km/h (average brisk walking speed) was chosen as it is recommended by the CDC as an adequate walking speed for health-promoting physical activity in adults. The speed of 8.1 km/h corresponded to the so-called average slow jogging speed and was chosen as it was expected to create less artifacts when re-cording EMG activity and to minimize the possibility of participants falling especially when running with 3 cm LLD (Lines 158-164).

  1. Reviewer’s comment on Line 134-135: The subjects wearing shoes before putting on the special rubber shoe (as shown in Figure 2). Is there a standardized criterion for the type of shoe used?

Authors’ response: There was no standardized criterion for the type of shoe used. Many studies in the past have used similar approaches to artificially increase leg length. This shoe was chosen based on the stability it can provide as it is suggested to be worn over a walking or running shoe to even up the length of the legs in individuals who are required to wear a walking boot as a treatment means. Appropriate modifications were made in the text to comply with reviewer’s comments (Lines 164-169).

  1. Reviewer’s comment on Line 149: Please specify HR.

Authors’ response: HR was specified according to the reviewer’s comment (Line 181).

  1. Reviewer’s comment on Line 213-217: Please provide a specific description of which variables were tested by single-factor ANOVA and two-factor ANOVA. In the context of the content describing the comparison of LLD (in four conditions) and lower limbs (in two conditions), a two-factor ANOVA should suffice. Furthermore, if a two-factor ANOVA yields an interaction effect, which indicates that the two factors are not independent and their combined effect differs from what would be expected if they were independent, how should this interaction be handled?

Authors’ response: A specific description of which variables were tested by single-factor ANOVA and two-factor ANOVA was provided according to the reviewer’s instructions. Please elaborate on the query on “…. how should this interaction be handled?” It is not clear what the reviewer wants us to comment on (Lines 244-249).

  1. Reviewer’s comment on Line 222-358: The overall results section appears rather complex, with significant differences marked, including p-values < .05, .01, and .001. It is essential to understand the statistical significance of p-values and only distinguish between "significant" and "non-significant" differences between variables. Overly detailed categorization labeling does not contribute to the interpretation of the results.

Authors’ response: Modifications were made in the Results section to comply with the reviewer’s instructions. For simplicity only interactions were reported for all variables measured (Lines 266-282, Lines 292-306, and Lines 319-346).

  1. Reviewer’s comment on Line 365-381: Are there any relevant research literature that can be compared with the results of this study?

Authors’ response: The paragraph between lines 416-430 contains an extensive review of the literature on scientific information related to spatiotemporal variables. A thorough literature review has been performed also on ground reaction forces and on EMG activity in individuals with LLD and is presented in lines 472-482 and in lines 532-542, respectively.

  1. Reviewer’s comment on Line 434-478: This passage primarily describes the results but does not provide sufficient discussion or explanations to help readers understand the significance of these findings and the underlying mechanisms.

Authors’ response: Lines 483-528 contain information on the underlying mechanism of GRFs exertion on the different plantar areas of both feet. The previous paragraph was modified according to the reviewer’s suggestions and has been amended and enriched with more information on the exertion of GRFs and their relationship with joint reaction forces. The clinical significance of the findings is presented in the separate paragraph named Clinical Implications.

  1. Reviewer’s comment on Line 512: what does the “longer limb(s)” mean?

Authors’ response:  “…longer limb…” was replaced with “…LLS…”  (Line 563).

  1. Reviewer’s comment on Line 479-516: The entire passage includes numerous comparisons with past research but lacks in-depth discussion.

Authors’ response: This paragraph included previous findings regarding kinematic changes in individuals with LLD that may have prevented EMG imbalance between sides. However, more information about the possible mechanism that these changes may account for the lack of muscle activation beyond the non-LLD state, was added as suggested by the reviewer (Lines 568-574).

  1. Reviewer’s comment on Line 518-520: The sentences may need to be restructured.

Authors’ response: The sentence was restructured according to the reviewer’s instructions (Lines 581-585).

  1. Reviewer’s comment on Line 542-564: The lack of a comparison of gender differences is a significant research limitation. Has this study employed experimental controls or statistical methods to reduce the potential result bias that might arise from gender differences?

Authors’ response: The purpose of the study was to investigate the differences between LLD and short and long legs in a sample of healthy individuals with minimal skeletal abnormalities in which it proved extremely difficult as all participants were active individuals participating in a variety of physical and sporting activities. Many of the volunteers were injured or had skeletal deviations above the limits considered normal for the purposes of this study. The repeated-measures ANOVA design was ultimately chosen because measuring the same subjects repeatedly could lead (i) to more efficient use of data, as it requires fewer participants to achieve the same statistical power compared to between-subjects designs and (ii) better control over confounding variables as participants serve as their own controls, minimizing the impact of individual differences that may confound the results. Further analysis is planned to investigate potential effects of gender and skeletal differences in artificially induced LLD.

  1. Reviewer’s comment on Line 565-579: The conclusion lacks clarity and comprehensiveness and requires a clearer summary and conclusion.

Authors’ response: The conclusion section was modified according to the reviewer’s instructions (Lines 630-649).

Reviewer 2 Report

Comments and Suggestions for Authors

Dear Authors,

the paper is interesting. I enclose a .pdf file of you manuscript along with some methological comments.

Best regards,

Comments on the Quality of English Language

 Minor editing of English language required

Reviewer 3 Report

Comments and Suggestions for Authors

This finding is clinically significant and interesting. I am satisfied with the quality of the work presented.

I do not have any substantial amendments to suggest.

But, one section is visually unclear.

The text in the figure 3-6 is small and should be improved with the overall layout.

Author Response

Dear Editor, Dear reviewer

Thank you for the effort you put into revising the manuscript thus giving us the opportunity to optimize its content. The comment was answered by the corresponding author as follows:

Reviewer’s comment: “The text in Figure 3-6 is small and should be improved with the overall layout”.

Authors’ response: The text in Figures 3-6 was modified according to the reviewer’s comment.

Round 2

Reviewer 1 Report

Comments and Suggestions for Authors

Line 242-249: In my perspective, this study investigates the bilateral differences in LLD sides (LLS and SLS) and the degree of LLD (0, 1, 2, 3). I would like to confirm that LLS and SLS seem to be independent variables. In this case, is a mixed-design ANOVA more suitable for this study? Furthermore, it should be described how to handle the main effects and simple main effects if there is an interaction between the two factors.

It is recommended to halve the values on the Y-axis in Figure 5 to emphasize the differences between the two sides. It is recommended to halve the values on the Y-axis in Figure 5 to emphasize the differences between the two sides. also, the fig 6 should adjust.

Author Response

Dear Editor, Dear reviewer

Thank you for the effort you put into revising the manuscript once again thus giving us the opportunity to optimize its content. All reviewer’s comments were answered by the corresponding author and are as follows:

Reviewer’s comment on Line 242-249: In my perspective, this study investigates the bilateral differences in LLD sides (LLS and SLS) and the degree of LLD (0, 1, 2, 3). I would like to confirm that LLS and SLS seem to be independent variables. In this case, is a mixed-design ANOVA more suitable for this study? Furthermore, it should be described how to handle the main effects and simple main effects if there is an interaction between the two factors.

Authors’ response:

  • Indeed, a mixed-design ANOVA was used to investigate the bilateral differences in LLD sides (LLS and SLS) and the degree of LLD (0, 1, 2, 3). Modifications were made in the manuscript to comply with the authors suggestions (Lines 92-98).
  • Reference to main effects was removed in the previous revised version of the manuscript to simplify the presentation of results. Furthermore, this was deemed necessary as most of the tested variables showed a significant interaction and reporting the main effects could be misleading to the reader. Interactions were interpreted by visual inspection of trends presented by values and bar graphs depending on how the variables were presented. Pairwise comparisons using a Bonferroni adjustment were then performed to identify potential significant differences between sides within each LLD and between LLDs for each side. This was decided to avoid over-analyzing the results, as a simple effects analysis would only provide the total effect of each independent variable at each level of the other independent variable. Certain modifications were made in the text to comply with the authors suggestions (Lines 250-253, 255-259).

Reviewer’s comment: It is recommended to halve the values on the Y-axis in Figure 5 to emphasize the differences between the two sides. It is recommended to halve the values on the Y-axis in Figure 5 to emphasize the differences between the two sides. also, the fig 6 should adjust.

Authors’ response:

  • Following the reviewer's suggestions, the Y-axis scale was modified to emphasize differences between sides. Ultimately, it turned out that the benefit of this modification was not what was expected. To emphasize the differences between the sides, the scale had to be adjusted in such a way that the differences between the sides appeared exaggerated at the lower speed compared to the other. The authors ultimately decided to keep the original graph presentation and for the information of the readers the differences between the sides regarding EMG activity of the muscles under investigation were reported with numerical values in the text (Lines 372-373)